# Synthesis and In Vitro Growth Inhibition of 2-Allylphenol Derivatives Against *Phythopthora cinnamomi* Rands

**DOI:** 10.3390/molecules24224196

**Published:** 2019-11-19

**Authors:** Andrés F. Olea, Luis Espinoza, Claudia Sedan, Mario Thomas, Rolando Martínez, Marco Mellado, Héctor Carrasco, Katy Díaz

**Affiliations:** 1Instituto de Ciencias Químicas Aplicadas, Facultad de Ingeniería, Universidad Autónoma de Chile, El Llano Subercaseaux 2801, Santiago CP 8900000, Chile; andres.olea@uautonoma.cl; 2Departamento de Química, Universidad Técnica Federico Santa María, Avenida España 1680, Valparaíso CP 2340000, Chile; luis.espinozac@usm.cl; 3Departamento de Ciencias Químicas, Facultad de Ciencias Exactas, Universidad Andrés Bello, Quillota 910, Viña del Mar CP 2520000, Chile; claudiasedan@hotmail.com (C.S.); m.thomaselgueda@gmail.com (M.T.);; 4Instituto de Química, Facultad de Ciencias, Pontificia Universidad Católica de Valparaíso, Avenida Universidad 330, Curauma, Valparaíso CP 2340000, Chile; marco.mellado@pucv.cl

**Keywords:** pest control, *Phytophthora cinnamomi*, fungicide, 2-allylphenol, structure–activity relationship

## Abstract

*Phytophthora cinnamomi* is a phytopathogen that causes extensive damage in different crops, and therefore, produces important economic losses all around the world. Chemical fungicides are a key factor for the control of this disease. However, ecological and environmental considerations, as well as the appearance of strains that are resistant to commercial fungicides, have prompted the quest for new antifungal agents which are of low ecological impact. In this work, a series of new 2-allylphenol derivatives was synthesized, and their structures were confirmed by FT-IR, NMR, and MS. Some of the synthesized compounds, more specifically nitro derivatives, exhibit strong growth inhibition of *P. cinnamomi* with EC_50_ as low as 10.0 µg/mL. This level of activity is similar to that exhibited by *METALAXYL MZ 58 WP*, a commonly-used commercial fungicide; therefore, these compounds might be of agricultural interest due to their potential use as fungicides against *P. cinnamomi*. The results indicate that this activity depends on the chemical structures of the 2-allylphenol derivatives, and that it is strongly enhanced in molecules where nitro and hydroxyl groups adopt a -para configuration. These effects are discussed in terms of the electronic distribution of the aromatic ring induced by substituent groups.

## 1. Introduction

Diseases of plants caused by species of the genus *Phytophthora* represent some of the most devastating crop diseases worldwide. All known species of *Phytophthora* are plant pathogens [1]. It has worldwide distribution and a host range approaching 5000 species [2,3]. In addition to causing substantial economic loss in agriculture, forestry, and horticulture, the inadvertent introduction of *Phytophthora cinnamomi* has had disastrous consequences for natural ecosystems and biodiversity [4]. The scale of *P. cinnamomi* distribution, its ability to survive for years in soil or symptomless plants, and the extent of plant susceptibility make the management of *P. cinnamomi* diseases challenging and difficult. Based on these facts, *P. cinnamomi* has been ranked in the top ten Oomycete plant pathogens [3].

In addition, *Phytophthora* shows important structural differences with respect to true fungi, which induce changes in the mechanism and efficacy of fungicides to control this phytopathogen. In other words, compounds that exhibit antifungal activity have no effect against oomycetes. The most commonly-used chemicals to control *P. cinnamomi* are metalaxyl or mefenoxam (phenylamides) and phosphite (a phosphonate and active ingredient in fosetyl-Al). However, the development of resistance to chemical fungicides have been detected, mainly in soils that have been treated for long periods with these antifungal compounds [5,6]. Thus, it is of great interest to find new chemical compounds that are active against these species and, hopefully, harmless for both people and the environment. From this point of view, the study of secondary metabolites and their hemisynthetic derivatives emerges as an interesting alternative to be explored. In this line, we have previously studied the antifungal activity of a series of phenols with side alkyl chains, such as geranylated phenols [7,8,9,10] and eugenol derivatives [11]. In these studies, it has been shown that fungicide activity depends on the nature, number, and position of substituents in the aromatic ring.

On the other hand, 2-allylphenol (2-AP) is a synthetic fungicide molecule that is structurally related to eugenol, and mimics the compound ginkgol found in gingko fruit (*Gingko biloba* L.). Recently, it has been shown that its fungicide activity against *Botrytis cinerea* is associated with increasing the cell respiration, i.e., mycelial oxygen uptake increases by 19.58% and 24.56% at 2-AP concentrations of 5 and 10 ppm, respectively [12]. Additionally, at 150 ppm, the oxygen consumption rate decreases by 77.49% in comparison with the control. These results indicated that at low concentrations, 2-AP increases cellular metabolism, whereas at high concentrations, it induces cell death [12]. Recently, the inhibition of mycelial growth of *Ramulispora cerealis*, *Pythium aphanidermatum*, *Valsa mali*, and *B. cinerea* by 2-AP derivatives, 2-(2-hydroxypropyl)phenol and 2-(3-hydroxypropyl)phenol has been assessed [13]. The results indicate that the chemical modification of the side allyl chain influences the antifungal activity, i.e., EC_50_ values of 2-(2-hydroxypropyl) were lower (1.0 to 23.5 ppm) than those measured for 2-allylphenol (8.2 to 48.8 ppm) [14].

Thus, these results suggest that 2-AP and its derivatives could be an interesting means by which to obtain new antifungal agents. We have previously shown that the antifungal activity of eugenol on *B. cinerea* is enhanced by the presence of nitro groups in the aromatic ring [15]. This effect was attributed to the increasing reactivity of the nitro derivative and production of reactive oxygen species (ROS) in an enzymatic-mediated process. Thus, herein we report on the synthesis of a series of 2-AP derivatives in which nitro groups were attached to the aromatic ring. The growth inhibition effect on *P. cinnamoni* was evaluated for all these compounds. The results indicate that the highest activity corresponds to nitro derivatives adopting a –para configuration of nitro and hydroxyl groups.

## 2. Results and Discussion

A series of nine new derivatives of 2-AP (**1**), compounds **2**–**10** in Figure 1, have been synthesized and characterized by spectroscopic methods.

The reactions used to synthesize these derivatives are depicted in Scheme 1. Briefly, compounds **2**, **4**, and **9** were obtained from 2-AP following standard reactions. Reaction yields of 96% and 67% were obtained for compounds **2** and **3**, respectively. The nitration of 2-AP leads to a mixture of compounds **4** and **9** with low yields (20% and 21%). To improve the reaction yield of compound **4**, an alternative synthetic procedure was used to get **4** with a 72% yield. Similarly, the nitration of **2** generates a mixture of **5** and **7** with low yields (4% and 7%). Methyl ethers **6** and **10** are obtained by the reaction of phenols **4** and **9** with dimethylsulfate with high reaction yields, i.e., 67% and 78% respectively. Compound **8** is obtained by the treatment of **7** with potassium carbonate in methanol (80% yield). Finally, nitration of **2** leads to nitroacetate derivatives **5** and **7**, with 4% and 7% reaction yields, respectively.

### Growth Inhibition of 2-AP and Its Derivatives Against Phytophthora cinnamomi In Vitro

The antifungal activities of 2-AP and its derivatives against *P. cinnamomi* were determined by measuring mycelial growth inhibition in the absence and presence of different concentrations of all the assayed compounds. The results indicate that the inhibitory effect increases with increasing concentration and depends on the chemical structure of 2-AP derivatives. A typical measurement is shown in Figure 2 for compound **9**.

The percentages of inhibition (%I) were calculated for each concentration, and EC_50_ values (concentration causing 50% inhibition of mycelial growth) were obtained from fitting data to a dose-response curve (see Experimental section). The EC_50_ values for all assayed compounds are given in Table 1.

Interestingly, 2-AP has been assayed against various fungal plant pathogens; its EC_50_ values range from 8 μg/mL (*Rhizoctonia cerealis*) to 49 μg/mL (*B. cinerea*) [14]. However, to the best of our knowledge, this is the first report of 2-AP activity against *P. cinnamomi*, and the EC_50_ value obtained in this work is three times higher than that measured for *B. cinerea*. This variability on antifungal activity suggests that the action mechanism of 2-AP changes from one fungal species to another. In addition, the results shown in Table 1 indicate that growth inhibition of *P. cinnamomi* (EC_50_ values) is clearly dependent on the chemical structure of 2-AP derivatives. For example, a two-fold increase in EC_50_ values is observed by acetylation of hydroxyl group (compare **1** and **2**). This activity decrease is even more important when the hydroxyl group is replaced by a methoxy group in compound **3**. On the other hand, the introduction of nitro groups in the aromatic ring of 2-AP brings about the opposite effect, depending on its relative position to the hydroxyl group, i.e., activity slightly decreases with substitution in the -ortho position (compares **1** and **4**), whereas it increases 15 times when a nitro group is attached in the -para position to the hydroxyl group (compares **1** and **8**). Interestingly, a similar effect is observed in nitro derivatives where the hydroxyl group has been replaced by acetyl or methoxy groups. For example, the completely inactive methoxy derivative (**3**) becomes slightly active by adding a nitro group in the -ortho position (**6**), but an even more drastic activity increase is obtained by moving the nitro group to the -para position (**10**). For acetyl derivatives, the -para isomer is one order of magnitude more active than the -ortho isomer (compares **5** and **7**). Thus, it seems that this structural configuration of 2-AP nitro derivatives enhances the antifungal activity, independently of the presence of a hydroxyl group in the -para position. These results suggest that the structural features of 2-AP derivatives determining their antifungal activity are mainly associated with the distribution of substituent groups in the aromatic ring, instead of the presence or absence of a particular group. Similar results were obtained for the inhibition of *P. cinnamomi* by a series of geranylated phenols [8,16]. Thus, it seems that the mechanism of action on this fungus involves the whole molecule and depends on the electronic distribution existing on the aromatic ring. To confirm this hypothesis, a study of the electronic properties of 2-AP derivatives was carried out. Maps of the molecular electronic potential were obtained after the optimization of chemical structure by molecular mechanics methods. Figure 3 shows charge distribution maps obtained for compounds **1**, **4**, and **8** (2-AP, ortho- and para- nitro derivatives of 2-AP, respectively).

The electrostatic potential map obtained for 2-AP (1) shows the existence of a positive charge area in the vicinity of carbon atoms C1 and C6, due to the presence of the hydroxyl group. For the *o*-nitro derivative (compound **4**) the introduction of a nitro group in C6 decreases the positive charge in this area (change from blue to green) due to an intra-molecular hydrogen bond and induces the appearance of a negative charge around oxygen of the nitro group. This drastic change in charge polarity induces a slight decrease on activity, i.e., EC_50_ increases from 155 to 180 µg/mL (compounds **1** and **4**, Table 1). Therefore, it seems that positively- or negatively-charged areas are required for these molecules to bind electrostatically to *P. cinnamomi*. On the other hand, for the most active derivative, the map shows both positive and negative surfaces located on opposite sides of the aromatic ring. As consequence, EC_50_ of 8 (*p*-nitro isomer) is more than one order of magnitude lower than that obtained for 4 (*o*-isomer). Thus, a polar surface involving the whole molecule seems to be a main structural requirement for activity.

Metalaxyl is a systemic fungicide which is highly selective to oomycetes that has no effect on respiration [17]. These results have been attributed to the alteration of biosynthesis or membrane structure caused by the presence of cellulose instead of chitin in the cell membrane [17]. Similarly, the antifungal activity of eugenol and its nitro-derivatives against *B. cinerea* has been explained by the ability of these compounds to interact with the cell wall of the fungus, spreading inside the hypha and interfering in the enzymatic activity responsible for the growth of the fungus [15,18]. This ability has been attributed to the molecular lipophilicity, measured by C*log*P (calculated logarithm of the partition coefficient of the molecule between octanol and aqueous phase), and the chemical reactivity of the nitro groups [19]. Thus, to determine whether the growth inhibition of *P. cinnamomi* by 2-AP derivatives follows a similar mechanism, the values of C*log*P were calculated for all 2-AP derivatives. The C*log*P values were in the range of 2.3 to 2.9 and were similar to those calculated for other antifungal agents [20]; however, no correlation between activity and C*log*P was found. These results suggest that activity against *P. cinnamomi* is determined mainly by the reactivity of nitro derivatives, which is closely related to the electronic distribution in the whole molecule.

## 3. Experimental Section

### 3.1. General Information

Unless otherwise stated, all chemical reagents purchased (Merck, Darmstadt, Germany or Aldrich, St. Louis, MO, USA) were of the highest commercially-available purity and were used without previous purification. IR spectra were obtained in a Thermo Scientific Nicolet Impact 6700 FT-IR spectrometer using KBr pellets or as thin films, frequencies are reported in cm^−1^. ^1^H- and ^13^C-NMR (DEPT 135 and DEPT 90) were performed on a Bruker Avance 400 Digital NMR spectrometer, operating at 400.1 MHz for ^1^H and 100.6 MHz for ^13^C. Spectra were recorded in CDCl_3_ solutions, and are referenced to the residual peaks of CHCl_3_, δ = 7.26 ppm and δ = 77.0 ppm for ^1^H and ^13^C, respectively. Chemical shifts are reported in δ ppm and coupling constants (*J*) are given in Hz. Chemical shifts are reported in δ (ppm downfield from the TMS resonance) and coupling constants (*J*) are given in Hz. GC–MS was carried out using a SHIMADZU GCMS-QP2010 instrument (Tokyo, Japan) using a 30 m × 0.25 mm id., 0.25 μm Rtx-5MS capillary column with helium as carrier gas to 1.61 mL/min. The column temperature program was 60 °C for 1 min at 5 °C/min, then increased to 285 °C for 2 min at 15 °C/min. Silica gel (Merck 200–300 mesh) was used for column chromatography (CC) and silica gel plates GF-254 for thin layer chromatography (TLC). TLC spots were detected by UV light and by heating after spraying with 25% H_2_SO_4_ in H_2_O.

### 3.2. Synthesis of 2-AP Derivatives

The synthetic pathways to obtain the nine new derivatives of 2-AP are shown in Scheme 1. General and specific information is given below. Pure compounds were obtained by CC, and new compounds were characterized and verified by spectroscopic methods.

#### 3.2.1. Methylation Reactions

Phenols **1**, **4**, and **9** were reacted with dimethyl sulfate to give methyl ethers **3**, **6**, and **10** by the following synthetic procedure. To a stirred solution of phenol in dry acetone (50–100 mL) was added potassium carbonate and dimethyl sulfate. The reaction was left to continue overnight under reflux. After this period, the complete disappearance of the starting product was confirmed by TLC (ethyl acetate/hexane; 1:3 by volume). The crude product of the reaction is diluted in acetone (30 mL) and water (50 mL), and then extracted with dichloromethane (3 × 50 mL), dried with anhydrous, and vacuum evaporated. The pure product was obtained by CC (ethyl acetate/hexane; 1:9 by volume).

#### 3.2.2. Nitration Procedure

Nitro derivatives **4**, **9**, and **5**, **7** were obtained from 2-AP and compound **2**, respectively, by reaction with sulfonitric mixture (2 mL of concentrated nitric acid, 6 mL of concentrated sulfuric acid and 2 mL of water). To a stirred and cooled (2 °C) solution of phenol in dichloromethane (10 mL) was carefully added the sulfonitric mixture (2 mL). The reaction was left to continue for 30 min and then interrupted by adding water (10 mL). The complete disappearance of the starting product was confirmed by TLC (ethyl acetate/hexane; 1:3 by volume). The organic layer was washed with water (3 × 10 mL) to eliminate the excess of acid, dried with anhydrous Na_2_SO_4_, filtered, and the solvent was evaporated at low pressure. Pure products were separated by CC (ethyl acetate/hexane; 1:9 by volume).

#### 3.2.3. Preparation of 2-Allylphenyl Acetate (**2**)

This compound was obtained by the acetylation of 2-AP. To a stirred solution of 2-AP (5.0 g, 37 mmol) in dichloromethane (10 mL) was added 4-*N*,*N*-dimethylaminopyridine (DMAP) (0.487 g, 3.98 mmol) and acetic anhydride (15.5 mL, 0.16 mol). The reaction proceeded for 2.5 h at room temperature. After this period, the complete disappearance of the starting product was confirmed by means of TLC (ethyl acetate/hexane; 1:3 by volume). The reaction was stopped by adding a 10% solution of potassium hydrogen sulfate (50 mL). The reaction crude product was extracted with dichloromethane, and the organic phase was washed with water (3 × 20 mL), dried with anhydrous Na_2_SO_4_, and vacuum evaporated. The pure product (yellow oil) was obtained by CC (ethyl acetate/hexane; 1:9 by volume). Compound **2** (6.31 g, 96% yield): IR (cm^−1^): 3076 (=C-H); 1756 (C=O); 1637 (C=C); 1367 (-CH_2_); 1197 (C-O); 1168 (C-O-C). ^1^H-NMR: 7.24–7.16 (2H, m, H-3 and H-5); 7.05–7.03 (2H, m, H-4 and H6); 5.95–5.65 (1H, m, H-2′); 5.09–5.05 (2H, m, H-3′); 3.31 (2H, d, *J* = 6.6 Hz, H-1′); 2.28 (3H, s, OCH_3_). ^13^C-NMR: 148.9 (C-1); 135.8 (C-2′); 131.8 (C-2); 130.3 (C-3); 127.3 (C-4); 126.0 (C-5); 122.3 (C-6); 116.1 (C-3′); 34.6 (C-1′); 20.8(CH_3_). EI-MS (+) *m*/*z* 176 [M+] (100%).

#### 3.2.4. Preparation of 1-Allyl-2-Methoxybenzene (**3**)

This compound was obtained by the methylation of 2-AP (1.0 g, 5.71 mmol) in the presence of dry acetone (120 mL), potassium carbonate (5.5 g, 39.8 mmol) and dimethyl sulfate (4.85 mL, 51.1 mmol). Compound **3** (yellow oil) was obtained by CC (0.735 g, 67% yield): IR (cm^−1^): 3080 (=C-H); 1768 (C=O); 1522 (C=C); 1344 (CH_3_); 1184 (C-O); 1166 (C-O-C). ^1^H-NMR: 7.25 (1H, dd, *J* = 8.8 and 7.5 Hz, H-3); 7.18 (1H, t, *J* = 9.0 Hz, H-5); 6.93 (1H, t, *J* = 7.4 Hz, H-4); 6.88 (1H, d, *J* = 8.2 Hz, H-6); 6.09–5.99 (1H, m, H-2′); 5.11–5.06 (2H, m, H-3′); 3.85 (3H, s, CH_3_CO); 3.42 (2H, d, *J* = 6.6 Hz, H-1′). ^13^C-NMR: 157.2 (C-1); 137.0 (C-2′); 129.7 (C-5); 128.6 (C-2); 127.3 (C-3); 120.5 (C-4); 115.3 (C-3′); 110.3 (C-6); 55.3 (OCH_3_); 34.2 (C-1′). EI-MS (+) *m*/*z* non-detected.

#### 3.2.5. Preparation of 2-Allyl-6-Nitrophenol (**4**) and 2-Allyl-4,6-Dinitrophenol (**9**)

The nitration of 2-AP (1.0 g, 7.42 mmol) was performed in dichloromethane (5 mL) with sulfonitric mixture (2 mL) at 2 °C. Two fractions were obtained by CC. Fraction I: Compound **4**; reddish oil (285 mg, 21% yield); Fraction II: Compound **9**; brown oil (342 mg, 20% yield). Compound **4**: IR (cm^−1^): 3201 (OH); 3085 (=C-H); 2977 (C-H); 1609 (C=C); 1539 (-NO_2_); 1448 (C=C); 1329 (N=O); 1249 (C-O-C). ^1^H-NMR: 10.90 (s, 1H, OH); 7.99 (1H, d, *J* = 8.6 Hz, H-5); 7.46 (1H, d, *J* = 7.3 Hz, H-3); 6.92 (1H, t, *J* = 8.0 Hz, H-4); 6.03–5.93 (1H, m, H-2′); 5.13–5.09 (2H, m, H-3′); 3.48 (2H, d, *J* = 6.5 Hz, H-1′).^13^C-NMR: 153.3 (C-6); 137.5 (C-3); 135.1 (C-2′); 131.4 (C-1); 123.1 (C-5); 119.5 (C-4); 112.5 (C-3); 116.8 (C-3′); 33.6 (C-1′). EI-MS (+) *m*/*z* 179 [M +] (100%) Compound **9**: IR (cm^−1^): 3218 (OH); 3105 (=C-H); 2921 (C-H); 1610 (C=C); 1331 (CH_2_); 1552 (-NO_2_); 1432 (N=O). ^1^H-NMR: 11.40 (1H, s, OH); 8.95 (1H, d, *J* = 2.7 Hz, H-5); 8.34 (1H, d, *J* = 2.2 Hz, H-3); 6.03–5.93 (1H, m, H-2′); 5.27–5.19 (2H, m, H-3′); 3.57 (2H, d, *J* = 6.6 Hz, H-1′). ^13^C-NMR: 157.3 (C-1); 139.7 (C-4); 133.8 (C-6); 133.2 (C-2′); 130.9 (C-3); 119.7 (C-5); 118.7 (C-3′); 33.6 (C-1′). EI-MS (+) *m*/*z* 224 [M+] (100%).

Because of the low reaction yield obtained with the previous reaction, the following alternative synthetic path was used to obtain compound **4**. First, 2-AP (5.0 g, 37 mmol) was dissolved in dichloromethane (100 mL) and added to a mixture containing potassium hydrogen sulfate (18.3 g, 0.134 mol), sodium nitrate (12.2 g, 0.144 mol), and wet silica (14.25 g, 50% *w*/*w*). The mixture was stirred under reflux for 24 h. The complete disappearance of the starting product was confirmed by TLC (ethyl acetate/hexane; 1:3 by volume). The solid product was filtered through silica and washed with dichloromethane; the solvent was evaporated in a vacuum to yield a reddish oil. The pure product (4.8 g, 72% yield) was obtained by CC (ethyl acetate/hexane; 1:9 by volume).

#### 3.2.6. Preparation of 2-Allyl-6-Nitrophenyl Acetate (**5**) and 2-Allyl-4-Nitrophenyl Acetate (**7**)

Nitration of compound **2** (1.0 g 5.71 mmol) was carried out in dichloromethane (8 mL) using sulfonitric mixture (2 mL) at 2 °C. Two fractions were obtained by CC. Fraction I: Compound **5**; yellow oil (47 mg, 4% yield); Fraction II: Compound **7**; reddish oil (87 mg, 7% yield). Compound **5**: IR (cm^−1^): 3080 (=C-H); 2930 (C-H); 1768 (C=O); 1522 (N-O); 1344 (N-O); 1184 (C-O); 1165 (C-O-C). ^1^H-NMR: 7.95 (1H, dd, *J* = 8.2 and 6.6 Hz, H-5); 7.54 (1H, dd, *J* = 7.7 and 6.4 Hz, H-3); 7.32 (1H, t, *J* = 15.8 Hz, H-4); 5.92–5.82 (1H, m, H-2′); 5.15–5.07 (2H, m, H-3′); 3.38 (2H, d, *J* = 6.6 Hz, H-1′); 2.37 (3H, s, CH_3_CO).^13^C-NMR: 168.3 (CO); 142.4 (C-1); 135.9 (C-6); 135.5 (C-3); 134.5 (C-2′); 121.0 (C-2); 117.5 (C-3′); 34.4 (C-1′); 20.7 (CH_3_CO). EI-MS (+) *m*/*z* 43 (CH_3_CO); 179 (C_9_H_8_NO_3_); 221 [M+] Compound **7**: IR (cm^−1^): 3085 (=C-H); 2915 (C-H); 1767(C=O); 1638 (C=C); 1521 (N-O); 1344 (N-O); 1183 (C-O); 1166 (C-O-C). ^1^H-NMR: 8.14–8.10 (2H, m, H-3 and H-5); 7.23 (1H, d, *J* = 8.8 Hz, H-6); 5.93–5.83 (1H, m, H-2′); 5.16 (1H, dd, *J* = 9.9 and 0.8 Hz H-3a′); 5.12 (1H, dd, *J* = 15.8 and 1.3 Hz, H-3′b); 3.37 (2H, d, *J* = 6.6 Hz, H-1′); 2.34 (3H, s, CH_3_CO) ^13^C-NMR: 168.2 (CO); 153.6 (C-4); 145.5 (C-1); 134.0 (C-2′); 133.8 (C-2); 125.6 (C-3); 123.3 (C-6); 122.8 (C-5); 117.7 (C-3′); 34.3(C-1′); 20.8 (CH_3_CO). EI-MS (+) *m*/*z* equal Compound **5**.

#### 3.2.7. Preparation of 1-Allyl-2-Methoxy-3-Nitrobenzene (**6**)

Compound **4** (0.5 g, 2.23 mmol) was methylated in dry acetone (45 mL) with potassium carbonate (2.5 g, 18.1 mmol) and dimethylsulfate (2 mL, 21 mmol). Compound **6** (reddish oil) was obtained by CC (418 mg, 78% yield). Compound **6**: IR (cm^−1^): 3085 (=C-H); 2937 (C-H); 1602 (NO_2_); 1523 (C=C); 1353 (N=O). ^1^H-NMR: 7.63 (1H, d, *J* = 1.6 Hz, H-4); 7.43 (1H, m, H-5); 7.15 (1H, t, *J* = 7.8 Hz, H-6); 6.00–5.90 (2H, m, H-2′); 5.15–5.06 (2H, m, H-3′); 3.88 (3H, s, CH_3_CO); 3.48 (2H, d, *J* = 7.7 Hz, H-1′). ^13^C-NMR: 151.4 (C-2); 144.2 (C-3); 136.4 (C-1);135.7 (C-2′); 135.0 (C-6); 123.8 (C-5); 123.4 (C-4); 117.0 (C-3′); 62.6 (OCH_3_); 33.5 (C-1′). EI-MS (+) *m*/*z* 193 [M+] (100%).

#### 3.2.8. Preparation of 2-Allyl-4-Nitrophenol (**8**)

A solution containing **7** (100 mg, 0.452 mmol) in methanol (5 mL) and K_2_CO_3_ (14 mg, 0.10 mmol) was stirred for 4 h at room temperature; then, HCl (0.1 M) was added to achieve pH = 2. The organic phase was extracted with CH_2_Cl_2_ (3 × 15 mL), washed with water, dried over Na_2_SO_4_, filtered, and evaporated. Subsequently, the mixture was purified by CC eluting with hexane + ethyl acetate mixtures of increasing polarity. Compound **8**; yellow oil (87 mg, 80% yield): IR (cm^−1^): 3353 (O-H); 2929 (C-H); 2361 (C=C); 1586 (NO_2_); 1493 (C=C); 1334 (N=O); 1280 (C-O). ^1^H-NMR: 8.06 (1H, d, *J* = 7.5 Hz, H-3); 8.05 (1H, dd, *J* = 7,5 Hz, H-5); 6.88 (1H, d, *J* = 9.1 Hz, H-6); 6.03–5.99 (2H, m, OH and H-2′); 5.26–5.19 (2H, m, H-3′); 3.46 (2H, d, *J* = 6.2 Hz, H-1′); ^13^C-NMR: 159.7 (C-1); 141.6 (C-4); 134.6 (C-2′); 126.4 (C-2); 126.3 (C-3); 124.3 (C-5); 118.0 (C-3′); 115.8 (C-6); 34.6 (C-1′). EI-MS (+) *m*/*z* 179 [M+] (100%).

#### 3.2.9. Preparation of 1-Allyl-6-Methoxy-3,5-Dinitrobenzene (**10**)

Compound **10** was obtained by methylation of 9 (530 mg, 2.37 mmol) with dimethylsulfate (1.6 mL, 16.9 mmol) in dry acetone (36 mL) and presence of potassium carbonate (1.63 g, 11.8 mmol). Compound **10** (reddish oil) was obtained by CC (380 mg, 67% yield). Compound **10**: IR (cm^−1^): 3094 (=C-H); 2924 (C-H); 1593 (C=C); 1530 (-NO_2_); 1477 (C=C); 1342 (N=O); 1264 (C-O-C). ^1^H-NMR: 8.57 (1H, d, *J* = 2.8 Hz, H-4); 8.30 (1H, d, *J* = 2.8 Hz, H-2); 6.00–5.90 (m, 1H, H-2′); 5.26 (1H, dd, *J* = 10.2 and 1.2 Hz, H-3a’); 5.17 (1H, d, *J* = 17.0 and 1.4 Hz, H-2b′); 3.98 (3H, s, OCH_3_); 3.55 (2H, d, *J* = 6.6 Hz, H-1′). ^13^C-NMR: 156.3 (C-6); 143.0 (C-3); 142.4 (C-1); 138.4 (C-2′); 133.9 (C-5); 128.8 (C-2); 119.5 (C-4); 63.0 (OCH_3_); 33.8 (C-3′). EI-MS (+) *m*/*z* 238 [M+] (100%).

### 3.3. Antifungal Activity of 2-Allylphenol and Derivatives Against Phytophthora Cinnamomi In Vitro

The antifungal activity of all studied compounds (**1**–**10**) against *P. cinnamomi* were tested according to a procedure previously reported [8]. Briefly, each compound was dissolved in an ethanol (5% *v*/*v*) solution and added to petri dishes (d = 50 mm) containing a potato dextrose agar medium (PDA, 5 mL) at 50 °C. The final concentrations of all the assayed compounds were: 10, 25, 50, 150, and 250 µg/mL. Solvent (1% ethanol) was taken as the negative control (C−), whereas METALAXIL MZ 48 WP (ANASAC, Santiago, Chile) commercial fungicide was used as positive control (C+) at the same concentrations and under the same conditions as those of the tested compounds. A mycelium agar disc (d = 4 mm) of the pathogen fungi was placed upside down in the center of PDA and incubated for six days at 23 °C in the dark. Colony diameters were measured when the colonies on the C− plates covered 100% of the agar surface. The inhibition percentages of mycelial growth for each compound were calculated by using a previously-described method [16]. Each treatment was replicated three times, and each assay was repeated twice. The effective concentration that inhibited mycelium growth by 50% (EC_50_) was obtained for each compound by fitting %I and concentration to a dose-response equation (see Figure 4). The fit analysis was performed using the Origin 8.0 software.

### 3.4. Computational Details

All 2-AP derivatives synthetized were optimized using Gaussian 09 and the DFT-B3LYP-6-31G (d,p) basis set, and verified by frequency calculations (without imaginary frequency) in gas phase calculus. After optimization, the electrostatic potential map of each compound was plotted. The lipophilicity index (C*Log*P) were obtained after molecular mechanic (MM) optimization using ChemDraw 3D 15.0.

## 4. Conclusions

In this work, a series of phenylpropanoids derived from 2-AP was synthetized and evaluated for its antifungal properties on mycelial growth of *P. cinnamomi*. The results show that some of these compounds are as effective as metalaxil, a commonly-used commercial fungicide, at inhibiting the mycelial growth of *P. cinnamomi*. The growth inhibition of the parent compound (2-AP) changes by attaching electron-acceptor groups to the aromatic ring, or by chemical modification of the hydroxyl group. In addition, this activity is strongly enhanced in those molecules where nitro and hydroxyl groups adopt a -para configuration. These effects are mainly attributed to changes in the electronic distribution on the aromatic ring induced by substituent groups.

It is worth mentioning that few known fungicides exhibit activity against *P. cinnamomi* at such low concentrations, which convert these molecules in potential precursors of useful fungicides for agricultural applications focused to control the infection caused for *P. cinnamomi*. However, further work is needed to increase the yields of reactions, leading to the most active compounds, and to determine the mechanism of action of these nitro derivatives.

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
