# Peer review of "Synthesis and In Vitro Growth Inhibition of 2-Allylphenol Derivatives Against Phythopthora cinnamomi Rands"

_molecules, 2019, doi:10.3390/molecules24224196_

Round 1

Reviewer 1 Report

Authors synthesized nine novel 2-allylphenol derivatives and studied their growth inhibition activity against a phytophathogenic fungi, Phythophora cinnamoni. The study looks well done and the results are of economic importance. As biologist, however, I can only partly judge the chemical part of the study. In my opinion, the manuscript can be accepted for publication after only minor revision:

line 77: activity correponds... line 275: give number of mmoles line 293: in vitro in lowercase letters line 295: give type of mixture ratio (v/v?) line 309: use points in axis labeling; this Fig. 4 is not mentioned in the text check References for a uniform use of lowercase and uppercase letters in names of papers; give species names in italics

Author Response

Reviewer 1

Authors synthesized nine novel 2-allylphenol derivatives and studied their growth inhibition activity against a phytophathogenic fungi, Phythophora cinnamoni. The study looks well done and the results are of economic importance. As biologist, however, I can only partly judge the chemical part of the study. In my opinion, the manuscript can be accepted for publication after only minor revision.

Some minor details:

1. line 77: activity corresponds....

Typo was corrected.

2. line 275: give number of mmoles.

We have included number of mmoles in the text.

3. line 293: in vitro in lowercase letters.

Done

4. line 295: give type of mixture ratio (v/v?).

The type of mixture was specified. It is %v/v

5. line 309: use points in axis labeling, this Fig. 4 is not mentioned in the text.

The format of X labeling was corrected. This Figure is mentioned in line 329 page 10.

6. check References for a uniform use of lowercase and uppercase letters in names of papers; give species names in italics.

The format in all references has been standardized.

Reviewer 2 Report

English language should be improved, in the text and in the title (I would suggest ommiting the word "activity". My main concern is that 9 compounds is not enough to establish a relevant SAR. Also, characterisation of compounds by IR and NMR is not complete, at least MS should be included. I would like to see the data about the purity of these compounds, eighter CHN analysis/HPLC purity. What about the human toxicity?                                                                                                                                                                                                                                                                                                                                                                                                                                                                                                                                                                                                                                                                                                                                                 

Author Response

Reviewer 2

English language should be improved, in the text and in the title (I would suggest omitting the word "activity".

English language was improved in all text. Additionally, we have eliminated the word “activity” from the title. However, in other part of the text “activity” is used instead of “growth inhibition” or to indicate “antifungal activity” of 2-AP derivatives, and in these cases, we have preferred to maintain this word.

My main concern is that 9 compounds is not enough to establish a relevant SAR.

We agree with the referee and many more compounds are required to get a SAR. By this reason we have simply emphasize the effect of nitro groups in growth inhibition of these compounds.

Also, characterization of compounds by IR and NMR is not complete, at least MS should be included.

Spectroscopic characterization of all compounds has been completed and MS spectra of all compounds, with exception of compound 3, are now included in the Experimental section. There was a non-identified problem in the MS detection of compound 3, but we believe that IR and NMR data provide enough information to assure the formation of this molecule.

I would like to see the data about the purity of these compounds, eighter CHN analysis/HPLC purity.

Unfortunately, we do not have the tools to measure purity, since currently there are no elementary analysis equipment working properly in Chile. However, we believe that spectroscopic characterization might be enough to assure that antifungal activity is mainly due to these compounds.

What about the human toxicity?

We believe that human toxicity is not a main issue in this work because we are presenting preliminary results on antifungal activity for these compounds.

Reviewer 3 Report

The current manuscript reports the synthesis of phenylpropanoids from 2-allylphenol with antifungal activity. synthetic procedures were narrated clearly, and antifungal activity is interesting.
However, there are some points require the attention of the authors as below;

1. All the derivatives are not new, nor employing novel synthetic methods.
2. Also, discussions about previous studies about the prepared compounds are not enough for the readership of the Journal.
3. Calls for the same compound are confusing. For example, there are three names for compound 1.

typos

line 148; 1 in Bold, please check other compounds numbers, should be in Bold.

Author Response

Reviewer 3

The current manuscript reports the synthesis of phenylpropanoids from 2-allylphenol with antifungal activity. synthetic procedures were narrated clearly, and antifungal activity is interesting.

However, there are some points require the attention of the authors as below; 

All the derivatives are not new, nor employing novel synthetic methods.

We agree with the reviewer in this point, but it is worth to mention that the main goal of this work is to evaluate the effect of chemical structure of 2-AP derivatives on the antifungal activity against P. cinnamomi, and no the synthesis or development of new synthesis tools.

Also, discussions about previous studies about the prepared compounds are not enough for the readership of the Journal.

This point is kind of confusing. All previous studies where the synthesized derivatives have been prepared are well documented and referenced properly. Thus, anyone interested on this particular issue could revise such references and go any further.

Calls for the same compound are confusing. For example, there are three names for compound 1.

We have corrected this situation and we have employed the term 2-AP throughout the text.

typos

line 148; 1 in Bold, please check other compounds numbers, should be in Bold.

Done, additionally we have checked compounds number in the text.

Round 2

Reviewer 2 Report

The authors have improved the manuscript. English language is fine, and only minor spelling sceck is required.

In the experimental part, please correct MS data, it is M+, not M+.

Still, some data about citotoxycity would be of great value. 

Reviewer 3 Report

Besides the previous comments about the non-novelty on chemical structure and synthetic methods, antifungal activity of the most potent entities in the manuscript is reported before. (Molecules 2012, 17, 1002)

One of the advances in this manuscript is the charge distribution map shown in figure 3. However, as in the manuscript, charge distribution is not enough to explain the change of activity through the derivatives.

Another advance is the calculation of ClogP, which is also not enough to explain the change of activity.

In this point of view, I am sorry to say that the current manuscript is not enough to be published in the Molecule.